

# Association of serum Metrnl levels and high-density lipoprotein cholesterol in patients with type 2 diabetes mellitus: a cross-sectional study

Chenxia Zhou[1],[*], Juli Zeng[2],[*], Xiangyu Gao[1], Da Chen[1], Qiugen Zhu[2], Bo Feng[1] and Jun Song[1]

[1] Department of Endocrinology, Shanghai East Hospital, Tongji University School of Medicine, Shanghai, Shanghai, China
[2] Department of Endocrinology, Shanghai East Hospital, Ji'An Hospital, Jiangxi, Ji'An, China
[*] These authors contributed equally to this work.

## ABSTRACT

**Purpose:** Meteorin-like (Metrnl) is a novel adipokine which is highly expressed in adipose tissue and has a beneficial effect on glucose and lipid metabolism. High density lipoprotein cholesterol (HDL-C) is well recognized to be inversely associated with cardiovascular events. However, the relationship between serum Metrnl levels and HDL-C in the type 2 diabetes mellitus (T2DM) remains unclear. Therefore, the present study aimed to evaluate the association of serum Metrnl with HDL-C levels in T2DM.

**Materials and Methods:** Eighty participants with T2DM were included in this cross-sectional study. They were divided into two groups according to HDL-C levels: Group1 (lower HDL-C group): HDL-C < 1.04 mmol/L; Group2 (higher HDL-C group): HDL-C ≥ 1.04 mmol/L. Serum Metrnl levels were measured by enzyme-linked immunosorbent assay (ELISA).

**Results:** As compared with lower HDL-C levels groups, serum Metrnl levels were significantly higher in the group with higher HDL-C. Binary logistic regression analysis showed serum Metrnl levels were positively associated with HDL-C group after adjustment with sex, age, body mass index (BMI), mean arterial pressure (MAP), fasting blood glucose (FPG), triglyceride (TG). Furthermore, serum Metrnl levels were inversely correlated with insulin resistance index (HOMA-IR). HDL-C levels were lowest in the group with the lowest Metrnl levels group and remained positively associated with Metrnl after adjustment for sex, age, BMI, TG, and HOMA-IR by using multivariate logistic regression analysis.

**Conclusion:** Serum Metrnl levels were positively associated with HDL-C levels in patients with T2DM.This suggests that increasing serum Metrnl levels maybe a candidate for improving lipid metabolism and preventing cardiovascular events in T2DM.

**Registry and the Registration No. of the Study/Trial:** The study was registered in the Chinese clinical trial registry (ChiCTR- 2100047148).

Corresponding authors
Bo Feng, fengbodfyy@tongji.edu.cn
Jun Song, songjuntj@tongji.edu.cn

## INTRODUCTION

Type 2 diabetes mellitus (T2DM) is a prominent public health concern worldwide, exhibiting profound morbidity and mortality rates. Among the numerous complications associated with T2DM, cardiovascular complications account for a substantial proportion of the mortality burden. Dyslipidemia, a well-established risk factor, plays a pivotal role in the heightened susceptibility of individuals with T2DM to cardiovascular diseases. The prevalence of dyslipidemia in T2DM patients ranges from 72% to 85% (*Schetz et al., 2019*; *Turner et al., 1998*), characterized by hypertriglyceridemia, elevated levels of low-density lipoprotein cholesterol (LDL-C), and diminished levels of high-density lipoprotein cholesterol (HDL-C). Accumulated evidence has unequivocally demonstrated a significant association between dyslipidemia in diabetes (DD) and an augmented risk of developing atherosclerotic cardiovascular disease (ASCVD).

Numerous large-scale epidemiological studies have provided compelling evidence on the inverse correlation between levels of HDL-C and the risk of coronary heart disease (CHD) (*Hwang et al., 2014*). Patients exhibiting low HDL-C levels are twice as likely to develop diabetes and more sensitive to suffer from cardiovascular complications, peripheral neuropathy, and diabetic nephropathy (*Ahmed et al., 2016*; *Wong et al., 2018*). In addition, it has been demonstrated that HDL boosts skeletal muscle glucose absorption (*Cochran et al., 2021*) and inhibit β-cell apoptosis in diabetes (*Wong et al., 2018*). Drew BG et al have demonstrated that HDL increases insulin sensitivity of peripheral tissues in T2DM (*Drew et al., 2012*). In addition to directly affecting glucose metabolism, HDL also plays an important role in transporting cholesterol from the arterial wall to the liver, protecting LDL from oxidation, promoting vasodilation, and reducing vascular endothelial cell inflammation (*Bonilha et al., 2021*; *Wong et al., 2018*). Increasing HDL levels may have a favorable effect on the natural progression of T2DM and prevent the onset of major cardiovascular events.

Adipose tissue is the largest secretory organ in the body, and several traditional adipokines have been found to participate in regulating HDL-C metabolisms. For example, adiponectin increases HDL biogenesis and decrease HDL catabolism (*Hafiane, Gasbarrino & Daskalopoulou, 2019*) by promoting cholesterol efflux. Clinical evidence also demonstrated adiponectin is positively correlated with HDL-C levels in T2DM patients (*Wang et al., 2020*). In addition, leptin is negatively correlated with HDL-C (*Wang et al., 2020*), and low-dose leptin treatment can reduce HDL-C levels in mice (*Silver, Jiang & Tall, 1999*).

Metrnl, an adipokine that was first characterized in 2014, exhibits expression in the subcutaneous adipose tissue of both rodents and humans. Emerging evidence has shown that Metrnl can induce the expression of genes related to thermogenesis in brown adipose tissue. Furthermore, Metrnl also demonstrates regulatory effects on the differentiation of adipocytes, and it shows potential in reducing inflammatory responses induced by lipid accumulation (*Jung et al., 2018*). Moreover, Metrnl regulates glucose metabolisms by enhancing glucose uptake and improving insulin resistance (*Cheng & Yu, 2022*). Several clinical studies have confirmed that have substantiated an inverse relationship between

serum Metrnl concentrations and various lipid parameters such as triglycerides (TG) and LDL-C levels among individuals with obesity (*Ding et al., 2022*; *Moradi et al., 2023*), T2DM (*Khajebishak et al., 2022*), coronary artery disease (*Giden & Yasak, 2023*; *Liu et al., 2019*).

T2DM is associated with high burden of chronic inflammation (*Basaran & Aktas, 2024*), Metrnl is also involved in inflammation (*Li et al., 2023*). Similarly, reduced HDL levels have been reported in diseases that characterized with chronic inflammation, such as prediabetes (*Balci et al., 2024*), chronic kidney disease (*Cheng et al., 2022*), and T2DM (*Bilgin et al., 2022*). Therefore, it is of interest to investigate the relationship between Metrnl level and HDL-C in T2DM patients. The primary objective of this study was to examine the potential correlation between serum Metrnl concentrations and HDL-C levels among patients diagnosed with T2DM.

## MATERIALS AND METHODS

### Study population

Between June 2022 and December 2022, a group of participants with T2DM were recruited from the Department of Endocrinology at the Ji'an Branch of Shanghai East Hospital for this cross-sectional study. The diagnosis of diabetes was made in accordance with the standards set by the World Health Organization in 2009. The exclusion criteria were type 1 diabetes, breastfeeding or pregnancy, hepatic or renal disease, thyroid dysfunction, cancer, open infection, anemia, and autoimmune disease. Ultimately, a total of 80 participants were included in the final evaluation. Every participant provided informed written consent, and the research procedure was ethically approved by Shanghai East Hospital's respective ethics committees ([2021] No. (023)). Furthermore, the study has been duly registered in the Chinese Clinical Trial Registry (ChiCTR-2100047148).

### Anthropometric and biochemical assessment

Weight and height were measured according to standard protocol. The body mass index (BMI) was calculated by dividing weight by height squared (kg/m$^2$). The waist-to-hip ratio (WHR) was calculated using the ratio of the waist circumference to the hip circumference. The formula for calculating the mean arterial pressure (MAP) was (systolic pressure + 2 × diastolic pressure)/3.

Sampling was performed by peripheral vein puncture of the forearm by trained nurses after a 8 h overnight fast. The following parameters were measured using an autoanalyzer (Cobas C702; Roche Diagnostics, Fukuoka, Japan): TG, total cholesterol (TC), HDL-C, LDL-C and fasting plasma glucose (FPG). The fasting insulin (FINS), 2-h postprandial insulin(2hINS), fasting C-peptide (FCP) and 2-h postprandial C peptide(2hCP) was detected by chemiluminescence (Cobas E602; Roche Diagnostics, Fukuoka, Japan). Glycosylated hemoglobin (HbA1c) was measured using high-performance liquid chromatography (Tokyo, Japan). Serum Metrnl and adiponectin levels were measured with ELISA kit (Shanghai Zhili Biological Company, Shanghai, China). Briefly, specimens, standards, and HRP-labeled detection antibodies were successively added to the coated microwells precoated with antibody, then incubated, and washed thoroughly. The substrate was used for color development, and finally the absorbance was measured at a

wavelength of 450 nm using a microplate reader to calculate the sample concentration. The formula for calculating the insulin resistance index (HOMA-IR) was FPG (mmol/L) × FINS (mIU/L)/22.5. The formula for calculating the quantitative insulin-sensitivity check index (QUICKI) was 1/[log (FINS) + log (FPG)].

## Grouping

In accordance with the *Joint Committee on the Revision of the Guidelines for the Prevention and Treatment of Dyslipidemia in Chinese Adults (2016)*, individuals were split into two groups based on their HDL-L levels: HDL-C < 1.04 mmol/L is the lower HDL group, whereas HDL-C ≥ 1.04 mmol/L is the higher HDL group. Based on the tertiles of the individuals' serum Metrnl levels, three groups were formed: Tertile1: below 31.290 ng/mL; Tertile2: between 31.290 and 37.776 ng/mL; Tertile3: above 37.776 ng/mL.

## Statistical analysis

All analyses were carried out using IBM SPSS 25.0 (IBM Corp., Armonk, New York, USA) and GraphPad Prism 8.0 (GraphPad, San Diego, CA, USA). When displaying data for continuous variables, the mean ± standard deviation or median (interquartile range) are used. Indicators of categorical data are expressed as percentages (%). Utilizing t-tests and one-way analysis for normally distributed continuous variables, Kruskal-Wallis tests for skewed continuous data, Bonferroni *post-hoc* tests for examining group differences, and $\chi^2$ tests for variables with categories were the methods employed for univariate analysis. Binary and multivariate logistic regression studies were performed to look at the relationship between HDL-C levels and differential clinical markers. $P < 0.05$ was chosen as the statistical significance cutoff point.

# RESULTS

## Serum Metrnl levels were higher in higher HDL-C group

Based on their HDL-C levels, participants were split into two groups (Group1: < 1.04 mmol/L; Group2: ≥ 1.04 mmol/L) (Table 1). Age, gender, and the duration of diabetes do not differ significantly in statistics between the two groups. The group with decreased HDL-C levels had higher levels of FPG, 2hPG, HOMA-IR, HbA1c, TG, TG: HDL ratio and lower QUICKI ($P < 0.05$). Patients with greater HDL-C groups had higher serum Metrnl levels ($P = 0.006$) and adiponectin levels ($P = 0.005$) (Fig. 1).

## Correlation of Serum Metrnl with HDL-C levels

Using binary logistic regression analysis, we explored the connection between serum Metrnl levels and HDL-C groups. According to Table 2, a noteworthy inverse relationship was found in an unadjusted logistic regression model (Model 1), (OR = 0.250, 95% CI: [0.080–0.781], $P = 0.017$), between lower serum Metrnl concentrations and higher HDL-C relative to the highest tertile. To put it another way, it shows that serum Metrnl concentrations and HDL-C levels are positively correlated. When age and sex were taken into account (Model 2), the link persisted (OR = 0.254, 95% CI: [0.080–0.805], $P = 0.020$). An increase in serum Metrnl was still positively correlated with an increase in HDL-C

**Table 1 Characteristic of the study population in different HDL-C groups.**

| Variables | Group 1 $N = 39$ | Group 2 $N = 41$ | *P* value |
|---|---|---|---|
| Age (years) | 55.03 ± 12.42 | 56.29 ± 10.90 | 0.629 |
| Sex (male, %) | 36 (92.3) | 33 (80.5) | 0.125 |
| BMI (kg/m$^2$) | 24.55 ± 3.25 | 23.26 ± 3.32 | 0.093 |
| WHR | 0.94 ± 0.04 | 0.92 ± 0.06 | 0.210 |
| Diabetes duration (years) | 5 (2, 10) | 5 (1, 9.5) | 0.431 |
| MAP (mmHg) | 99.05 ± 11.39 | 96.76 ± 13.59 | 0.418 |
| FPG (mmol/L) | 10.29 (7.87, 11.94) | 7.84 (6.01, 9.51) | 0.003 |
| 2hPG (mmol/L) | 15.42 ± 4.94 | 12.82 ± 5.71 | 0.035 |
| FINS (U/mL) | 8.41 (4.10, 12.48) | 5.67 (2.85, 12.08) | 0.127 |
| 2hINS (U/mL) | 29.51 (21.30, 45.19) | 22.54 (12.36, 44.38) | 0.122 |
| FCP (ng/mL) | 2.27 (1.58, 3.70) | 1.91 (1.15, 3.15) | 0.125 |
| 2 h CP (ng/mL) | 4.90 (3.03, 6.96) | 4.27 (2.28, 6.84) | 0.436 |
| HOMA-IR (>1 (n)%) | 34 (89.5) | 27 (71.1) | 0.041 |
| HbAlc (%) | 9.25 (7.65, 11.95) | 8.10 (6.63, 9.88) | 0.027 |
| TC (mmol/L) | 4.90 ± 1.57 | 4.49 ± 1.05 | 0.181 |
| TG (mmol/L) | 3.23 (1.73, 6.05) | 1.18 (0.83, 1.63) | <0.001 |
| LDL-C (mmol/L) | 2.25 ± 0.86 | 2.56 ± 0.86 | 0.122 |
| Metrnl (ng/mL) | 31.29 (29.80, 37.88) | 35.13 (32.22, 47.52) | 0.006 |
| Adiponectin (ng/mL) | 19.23 ± 3.75 | 34.91 ± 33.93 | 0.005 |
| TG: HDL ratio | 6.37 (4.04, 8.71) | 1.30 (0.81, 1.79) | <0.001 |
| QUICKI | 0.55 (0.51, 0.59) | 0.69 (0.60, 0.78) | 0.015 |

**Note:**
Data are presented as mean ± standard deviation, median (interquartile range), or n (%). BMI, body mass index; WHR, waist-to-hip ratio; MAP, mean arterial pressure; FPG, fasting plasma glucose; 2hPG, postprandial 2-h plasma glucose; FINS, fasting insulin; 2hINS, 2-h postprandial insulin; FCP, fasting C-peptide; 2hCP, 2-h postprandial C peptide; HOMA-IR, insulin resistance index; HbA1c, glycosylated hemoglobin A1c; TC, total cholesterol; TG, triglycerides; LDL-C, low-density lipoprotein cholesterol; HDL-C, high-density lipoprotein cholesterol; QUICKI, quantitative insulin-sensitivity check index.

levels in Model 3, even after additional adjustments for sex, age, BMI, MAP, FPG, and TG (OR = 0.195, 95% CI: [0.046–0.815], *P* = 0.025).

Recent studies indicated Metrnl might promote cholesterol efflux *in vitro*. However, the association between Metrnl and HDL-C remains unclear. So, we performed multiple linear regression, by using continuous HDL-C data as the dependent variable and continuous TG, 2hPG, BMI, Metrnl as the independent variable, to investigate the correlation between HDL-C and serum Metrnl levels. Table 3 demonstrates a positive correlation (*P* < 0.01) with serum Metrnl levels and HDL-C, while a negative correlation (*P* < 0.01) was observed between HDL-C levels and TG, 2hPG, and BMI.

So as to investigate the connection between Metrnl grouping and Serum HDL-C levels further, each participant was separated into three groups (Table 4) based on the tertiles of their Metrnl levels: below 31.290 ng/mL for Tertile 1; between 31.290 and 37.776 ng/mL for Tertile 2; and above 37.776 ng/mL for Tertile 3. From the results, those with the lowest Metrnl concentrations had greater HOMA-IR (*P* = 0.035) but lower HDL-C levels

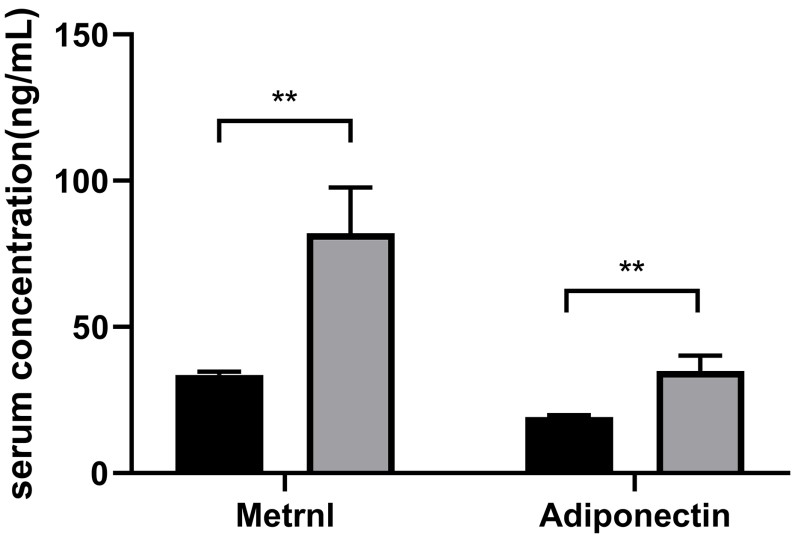

**Figure 1 Comparison of the Metrnl and adiponectin levels in different HDL-C groups.** Participants were divided into two groups according to their HDL-C levels (Group1: HDL-C < 1.04 mmol/L; Group2: HDL-C ≥ 1.04 mmol/L) (**$p < 0.01$).

**Table 2 Binary logistic regression of the relationship between serum Metrnl level and HDL-C group.**

|  | OR (95% CI) | | | *P* for trend |
|---|---|---|---|---|
|  | Tertile3 | Tertile2 | Tertile1 |  |
| Model1 | 1.00 (Ref) | 1.181 [0.381–3.655], 0.773 | 0.250 [0.080–0.781], 0.017 | 0.015 |
| Model2 | 1.00 (Ref) | 1.184 [0.377–3.722], 0.772 | 0.254 [0.080–0.805], 0.020 | 0.018 |
| Model3 | 1.00 (Ref) | 1.172 [0.252–5.450], 0.840 | 0.195 [0.046–0.815], 0.025 | 0.024 |

Notes:
Model1: unadjusted.
Model2: adjusted for sex, age.
Model3: adjusted for sex, age, BMI, MAP, FPG, TG.
BMI, body mass index; MAP, mean arterial pressure; FPG, fasting plasma glucose; TG, triglycerides.

**Table 3 The relationship between serum Metrnl level and HDL-C levels (multivariable linear regression).**

| Variable | Beta | SE | T | *P* value | B 95% CI |
|---|---|---|---|---|---|
| TG | −0.022 | 0.007 | −3.124 | 0.003 | [−0.036 to 0.008] |
| 2hPG | −0.022 | 0.005 | −4.208 | <0.001 | [−0.033 to 0.012] |
| BMI | −0.03 | 0.009 | −3.472 | 0.001 | [−0.047 to 0.013] |
| Metrnl | 0.001 | <0.001 | 2.812 | 0.007 | [0–0.002] |

Note:
TG, triglycerides; 2hPG, postprandial 2-h plasma glucose; BMI, body mass index.

($P = 0.019$) (Table 4 and Fig. 2). Serum HDL-C level was positively correlated with Metrnl, according to univariate ordinal logistic regression analysis (OR = 0.35, 95% CI: [0.15–1.25], $P = 0.013$). Even after accounting for sex and age (Model 2), this connection persisted (OR = 0.35, 95% CI: [0.15–1.22], $P = 0.016$). Following correcting for sex, age, BMI, TG, and HOMA-IR in model 3, there was still a significant correlation between

**Table 4 Characteristic of the study population in different Metrnl groups.**

| Variables | Tertile1 N = 28 | Tertile2 N = 26 | Tertile3 N = 26 | P value |
|---|---|---|---|---|
| Age (years) | 54.79 ± 12.741 | 56.27 ± 10.513 | 56.04 ± 11.780 | 0.882 |
| Sex (male,%) | 25 (89.3) | 22 (84.6) | 23 (85.2) | 0.849 |
| BMI (kg/m2) | 24.12 ± 3.01 | 24.03 ± 2.40 | 23.39 ± 4.39 | 0.706 |
| WHR | 0.93 ± 0.04 | 0.95 ± 0.04 | 0.91 ± 0.06 | 0.078 |
| Diabetes duration (years) | 5.00 (2.25, 10.00) | 5.00 (0.00, 10.25) | 6.00 (3.00, 10.00) | 0.549 |
| MAP (mmHg) | 97.19 ± 10.80 | 98.27 ± 12.97 | 97.60 ± 14.31 | 0.952 |
| FPG (mmol/L) | 9.86 (6.88, 11.49) | 8.28 (6.78, 11.49) | 8.65 (6.01, 10.98) | 0.778 |
| 2hPG (mmol/L) | 13.93 ± 3.86 | 14.45 ± 5.89 | 13.99 ± 6.46 | 0.934 |
| FINS (U/mL) | 7.18 (4.45, 14.07) | 7.71 (3.36, 11.15) | 6.18 (2.89, 12.32) | 0.251 |
| 2hINS (U/mL) | 29.17 (15.49, 39.00) | 26.07 (16.89, 48.33) | 25.08 (12.28, 47.89) | 0.996 |
| FCP (ng/mL) | 2.70 ± 1.44 | 2.30 ± 1.18 | 2.10 ± 1.30 | 0.234 |
| 2hCP (ng/mL) | 5.08 (3.41, 6.77) | 4.35 (2.55, 6.17) | 4.64 (1.75, 7.59) | 0.786 |
| HOMA-IR (>1 (n)%) | 25 (96.2) | 18 (75.0) | 19 (70.4) | 0.035 |
| HbAlc (%) | 8.75 ± 2.24 | 9.66 ± 2.94 | 9.16 ± 2.52 | 0.457 |
| TC (mmol/L) | 4.73 ± 1.33 | 4.60 ± 1.16 | 4.75 ± 1.55 | 0.911 |
| TG (mmol/L) | 2.57 (1.33, 3.61) | 1.39 (0.98, 3.08) | 1.37 (0.79, 3.36) | 0.070 |
| LDL-C (mmol/L) | 2.28 ± 0.90 | 2.51 ± 0.81 | 2.44 ± 0.92 | 0.604 |
| HDL-C (mmol/L) | 0.96 ± 0.23 | 1.15 ± 0.32 | 1.22 ± 0.44 | 0.019 |

Note:
Data are presented as mean ± standard deviation, median (interquartile range), or $n$ (%). BMI, body mass index; WHR, waist-to-hip ratio; MAP, mean arterial pressure; FPG, fasting plasma glucose; 2hPG, postprandial 2-h plasma glucose; FINS, fasting insulin; 2hINS, 2-h postprandial insulin; FCP, fasting C-peptide; 2hCP, 2-h postprandial C peptide; HOMA-IR, insulin resistance index; HbA1c, glycosylated hemoglobin A1c; TC, total cholesterol; TG, triglycerides; LDL-C, low-density lipoprotein cholesterol; HDL-C, high-density lipoprotein cholesterol.

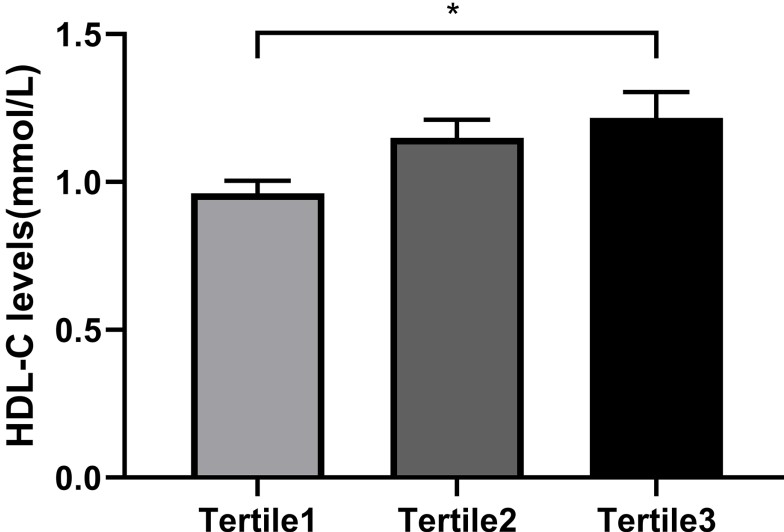

**Figure 2 Comparison of the HDL-C levels in different Metrnl groups.** All the participants were divided into three groups according to the tertiles of Metrnl levels: Tertile1: Metrnl < 31.290 ng/mL; Tertile2: 31.290 ≤ Metrnl ≤ 37.776 ng/mL; Tertile3: Metrnl > 37.776 ng/mL. (*$p < 0.05$).

**Table 5 The relationship between serum HDL level and Metrnl (multivariate ordinal logistic regression).**

| | OR (95% CI) | | P |
|---|---|---|---|
| | Group 2 | Group 1 | |
| Model1 | 1 | 0.35 [0.15–1.25] | 0.013 |
| Model2 | 1 | 0.35 [0.15–1.22] | 0.016 |
| Model3 | 1 | 0.82 [0.09–1.04] | 0.043 |

Notes:
Model1: unadjusted.
Model2: adjusted for sex, age.
Model3: adjusted for sex, age, BMI, TG, HOMA-IR.
BMI, body mass index; TG, triglycerides; HOMA-IR, insulin resistance index.

elevated Metrnl levels and serum HDL-C levels (OR = 0.82, 95% CI: [0.09–1.04], $P$ = 0.043) (Table 5). Moreover, we also analyzed the relationship between Metrnl levels and adiponectin levels in T2DM patients. Spearman correlation analysis showed that adiponectin level was positively correlated with Metrnl level (r = 0.369, $P$ = 0.001). This positive association persisted with univariate ordinal logistic regression of serum adiponectin with Metrnl tertiles as the outcome variable (OR = 0.02–0.15, $P$ = 0.011) (Model 1), after adjustment for age and sex (OR = 0.02–1.62, $P$ = 0.011) (Model 2), and then after adjustment for serum HDL levels (OR = 0.02–0.17, $P$ = 0.020) (Model 3) (Table S1). QUICKI and the triglyceride: HDL ratio are two other important measures to assess insulin sensitivity and insulin resistance. Spearman correlation analysis was used to examine its association with Metrnl levels and found that TG: HDL ratio was negatively correlated with Metrnl (r = −0.313, $P$ = 0.005), while QUICKI was not significantly associated with Metrnl levels (r = 0.156, $P$ = 0.176).

# DISCUSSION

In T2DM patients, our research revealed a beneficial correlation between blood concentrations of Metrnl and HDL-C, which may provide an objective for the prognosis and prevention of diabetic cardiovascular disease.

Dyslipidemia is widely recognized as the most important feature of type 2 diabetes and a major risk factor for the development, progression and prognosis of T2DM-related cardiovascular events. Increased TG, decreased HDL-C and elevated LDL-C are the typical hallmarks of diabetic dyslipidemia (*Turner et al., 1998*). The UK Prospective Diabetes Study (UKPDS) identified dyslipidemia as the primary underlying marker for ASCVD in people with T2DM. Our research found that 40.7% of people with T2DM had dyslipidemia (high TG, high TC, high LDL-C or low HDL-C, more than two of which), with low HDL-C accounting for 49.4% of cases.

Decreased HDL-C is one of the typical manifestations of dyslipidemia in the T2DM state. For a long time, HDL-C has been known as "good cholesterol" because of its negative association with cardiovascular events. This effect of HDL-C is attributed to its cholesterol efflux and anti-inflammatory effects in macrophages and vascular endothelial cells

(*Rohatgi et al., 2021*). HDL in human plasma is mainly composed of several different subgroups. Although these subgroups are functionally heterogeneous, most HDL contains apolipoprotein A-1 (ApoA-1) (*Rosenson et al., 2012*). ATP-binding cholesterol transporters ABCG1 and ABCA1 export cholesterol from macrophages in the vessel wall to HDL and ApoA-1, reducing lipid deposition and preventing foam cell formation, thereby attenuating atherosclerosis (*Ouimet, Barrett & Fisher, 2019*). This mechanism is an important basis for the CV protective function of HDL. In addition, HDL suppressed inflammation by enhancing the expression of activating transcription factor 3 (ATF3) in macrophages and endothelial cells (*De Nardo et al., 2014*). HDL-mediated cholesterol efflux also inhibits Toll-like receptor 4 (TLR4) downstream inflammatory signaling through ABCG1 and ABCA1, thereby inhibiting the NF-κB response in endothelial cells and macrophages (*Groenen et al., 2021*; *Rosenson et al., 2012*). In addition to its antiatherogenic properties, HDL has the latent capability to improve glycemic control in mouse models of diabetes by boosting pancreatic β-cell function and increasing insulin sensitivity (*Manandhar, Cochran & Rye, 2020*). Our research found that the group with decreased HDL-C levels had higher TG:HDL ratio and lower QUICKI, indicating the possible role of HDL-C in improving insulin resistance.

Although recent studies have shown that higher level of HDL-C is not better (*Baliga, Yang & Bossone, 2022*; *Liu et al., 2022*), elevated risk of CAD is associated with low levels of HDL-C, which are clinically important predictors of cardiovascular events (*Rohatgi et al., 2021*). Numerous extensive epidemiological investigations have verified an opposite relationship between ASCVD risk and HDL-C (*Gordon et al., 1989*; *Mazidi, Mikhailidis & Banach, 2019*). A prospective study involving 3,837 patients with vascular disease confirmed that low HDL-C levels were positively associated with cardiovascular risk independently of the use of lipid-lowering drugs and LDL-C levels (*Hajer et al., 2009*). Cohort studies have shown that HDL-C is an independent predictor of coronary events compared with triglyceride and LDL-C, the 0.026 mmol/L increase from baseline was associated with a 1.3% reduction in major cardiovascular risk (*Laitinen, Manthena & Webb, 2010*). Below 40 mg/dL, there was a linear inverse association between HDL-C and ASCVD mortality (*Liu et al., 2022*). A clinical retrospective analysis showed that HDL < 1.04 mmol/L is independent risk factors for carotid intima-media thickening in patients with T2DM (*Chen, Liang & Liang, 2021*) of 648 Spanish patients with an acute coronary syndrome, 367 (56.6%) had low HDL-C (HDL-C < 1.04 mmol/L) (*Pintó et al., 2010*), In contrast, a 20-year follow-up study in Japanese subjects showed that high serum HDL-C levels, at least as high as 2.06 mmol/L, were protective against coronary heart disease (*Hirata et al., 2016*). Therefore, in our study we defined HDL-C < 1.04 mmol/L as the lower HDL-C group.

Multiple adipokines participate in the regulation of HDL-C metabolisms, according to prior investigations. Adiponectin regulate HDL biogenesis by promoting ABCA1-dependent cholesterol efflux and activating peroxisome proliferator-activated receptor γ(PPAR-γ)/Liver X Receptor α (LXR-α) signaling pathway in macrophages (*Hafiane, Gasbarrino & Daskalopoulou, 2019*). Adiponectin can significantly increase HDL-C level

in patients with type 2 diabetes. Our results also showed that HDL-C level is positively correlated with Adiponectin level in T2DM patients. Leptin increases cholesterol acyltransferase 1 (ACAT-1) transcription in human monocyte-derived macrophages by upregulating Janus kinase-2 (JAK2) and phosphoinositide 3-kinase (PI3K), thereby reducing cholesterol efflux (*Hongo et al., 2009*). A clinical studies showed that serum leptin levels were independently associated with HDL-ApoA-I in 288 Mexican Americans (*Rainwater et al., 1997*). In patients with type 2 diabetes, HDL-C is positively correlated with serum adiponectin and negatively correlated with leptin (*Wang et al., 2020*).

Metrnl, a novel adipokine located in the 11Qe2 region of human and mouse chromosomes, is abundantly expressed in rodent and human subcutaneous adipose tissues, and is also highly expressed in skin and mucosal barrier tissues (*Zheng et al., 2016*). In previous studies, Metrnl has been demonstrated to inhibit steatosis, promote lipid metabolism, and ultimately improve insulin resistance in adipocyte through PPARγ pathway *via* an autocrine or paracrine mechanism (*Li et al., 2015*). Furthermore, Metrnl inhibited the beta-cell apoptosis induced by high levels of blood glucose (*Hu, Wang & Sun, 2021*) in T2DM mice. Injecting Metrnl retards the onset of diabetes in non-obese diabetic mice (*Yao et al., 2021*). Numerous investigations contend that serum Metrnl levels are linked with inflammation. Furthermore, inflammatory components are heightened in umbilical cord endothelial cells deficient in Metrnl. However, Metrnl improves inflammation triggered by LPS in endothelium *via* AMPK and PPAR-γ pathways (*Jung et al., 2021*). Other studies have found that Metrnl is decreased in patients with AS and coronary artery disease, and it is negatively correlated with endothelial function indicators (*Cai et al., 2022*; *El-Ashmawy et al., 2019*; *Fadaei et al., 2020*). A study showed an inverse relation between reduced Metrnl levels and the severity of CAD (*Liu et al., 2019*), indicating the protective role of Metrnl in preventing diabetes ASCVD. Adiponectin is a well-recognized adipokine with cardiovascular protective effects (*Hafiane, Gasbarrino & Daskalopoulou, 2019*). The positive correlation between adiponectin and Metrnl levels was also found in our study, further suggesting the protective effect of Metrnl on cardiovascular events. Due to the role of Metrnl in regulating glucose and lipid metabolism, several clinical studies have focused on influence on lipid metabolism. The study showed a negative correlation between serum Metrnl levels and lipid parameters such as total and low-density lipoprotein cholesterol. In individuals experiencing overweight or obesity, serum Metrnl levels are negatively associated with triglycerides, total cholesterol, and low-density lipoprotein cholesterol (*Ding et al., 2022*). In the context of type 2 diabetes, studies have demonstrated that Metrnl is adversely associated with LDL and TG (*Khajebishak et al., 2022*). Further studies have shown that Metrnl is involved in the HDL-C regulation. A preclinical study showed that serum HDL-C levels were decreased by 24% in high-fat-fed liver-specific Metrnl knockout mice compared to non-knockout mice (*Qi et al., 2020*). However, the association of Metrnl and HDL-C in diabetes patients has not been investigated. Our findings showed that serum Metrnl levels was positively associated with HDL-C in T2DM patients. We further compared the HDL-C levels with different Metrnl levels in T2DM patients. Similarly, it showed that individuals with the

lowest Metrnl concentrations had lower HDL-C levels. Univariate ordinal logistic regression analysis serum HDL-C level was significantly positively associated with Metrnl. After adjustment for confounding factors, the correlation remains. Given the reduced HDL-C level in liver-specific Metrnl knockout mice but no changes in lipid parameters in global Metrnl knockout mice (*Qi et al., 2020*), we hypothesized that Metrnl might regulate lipid metabolism by liver-related mechanisms. However, the specific mechanisms still need to be further explored. Consistent with the previous study (*Jung et al., 2018*; *Li et al., 2015*), we also found a significant difference in HOMA-IR in different Metrnl groups, lower level of Metrnl exacerbates insulin resistance. In addition, TG:HDL ratio, another important indicator of insulin resistance (*Baneu et al., 2024*), was also negatively correlated with Metrnl level. As previously mentioned, one possible mechanism is to reduce the insulin resistance by PPARγ pathway (*Li et al., 2015*). Further studies will be investigated on the relationship between Metrnl and insulin resistance.

There were several limitations in the current study. Firstly, it was a cross-sectional study without follow-up, which means that more work is needed to consolidate the findings. Secondly, the sample was limited to Chinese participants and therefore may lack generalizability. Furthermore, further investigations are necessary to elucidate the mechanisms of Metrnl in regulating HDL metabolism.

## CONCLUSION

Conclusively, the findings of this study demonstrate a positive correlated relationship between Metrnl and HDL-C levels in T2DM, indicating a candidate for improving lipid metabolism and preventing cardiovascular events in T2DM.

### Funding

This work was supported by the National Natural Science Foundation of China (82070842), the Clinical Research Fund of Shanghai Municipal Commission of Health (202040136), the China Diabetes Young Scientific Talent Research Project (2018-N-01), and the Jiangxi Health Commission Science and Technology Plan Project (202212838, 202212852). The funders had no role in study design, data collection and analysis, decision to publish, or preparation of the manuscript.

### Grant Disclosures

The following grant information was disclosed by the authors:
National Natural Science Foundation of China: 82070842.
Shanghai Municipal Commission of Health: 202040136.
China Diabetes Young Scientific Talent Research: 2018-N-01.
Jiangxi Health Commission Science and Technology: 202212838, 202212852.

### Competing Interests

The authors declare that they have no competing interests.

## Author Contributions

- Chenxia Zhou conceived and designed the experiments, performed the experiments, analyzed the data, prepared figures and/or tables, authored or reviewed drafts of the article, and approved the final draft.
- Juli Zeng conceived and designed the experiments, performed the experiments, analyzed the data, prepared figures and/or tables, and approved the final draft.
- Xiangyu Gao performed the experiments, analyzed the data, authored or reviewed drafts of the article, and approved the final draft.
- Da Chen performed the experiments, authored or reviewed drafts of the article, and approved the final draft.
- Qiugen Zhu analyzed the data, authored or reviewed drafts of the article, and approved the final draft.
- Bo Feng conceived and designed the experiments, authored or reviewed drafts of the article, and approved the final draft.
- Jun Song conceived and designed the experiments, authored or reviewed drafts of the article, and approved the final draft.

## Human Ethics

The following information was supplied relating to ethical approvals (*i.e.*, approving body and any reference numbers):

The ethics committees of Shanghai East Hospital.

## Data Availability

The original measurements are available in the Supplemental File.

## Supplemental Information

Supplemental information for this article can be found online at http://dx.doi.org/10.7717/peerj.18264#supplemental-information.

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
