# Peer review of "Association of serum Metrnl levels and high-density lipoprotein cholesterol in patients with type 2 diabetes mellitus: a cross-sectional study"

_PeerJ, doi:10.7717/peerj.18264_

## Round 0.1 · original submission · Minor Revisions

Please revise your manuscript and submit it with your written responses to each of the reviewers' comments.

Yours,

Yoshi

Prof. Yoshinori Marunaka, M.D., Ph.D.

Reviewer 1 ·

Basic reporting

Authors studied metrnl and HDL in patients with type 2 DM. Background data must be improved in this section since the rationale is missing. Type 2 DM is associated with high burden of chronic inflammation (AIMS Medical Science, 2024;11(1): 47–57. DOI: 10.3934/medsci.2024004). Meteorin like (Metrnl) protein is also involved in inflammation (Frontiers in immunology, 2023, 14: 1098570.). Similarly, reduced HDL levels have been reported in diseases that characterized with chronic inflammation, such as prediabetes (Bratisl Med J 2024;125(3): 145–148. DOI: 10.4149/BLL_2023_130), chronic kidney disease (BMJ open, 2022, 12.12: e066243.), and type 2 DM (Acta facultatis medicae Naissensis 2022; 39(1):66-73. DOI: 10.5937/afmnai39-33239). Hence, studying metrnl's association with hdl in diabetic subjects makes sense.

Experimental design

They retrospectively analyzed the data of the participants. Metrnl levels were compared in low HDL and high HDL groups.

Validity of the findings

Metrnl levels were higher in high HDL group compared to those in low HDL group. Conclusions are supported by the data.

Additional comments

Avoid abbreviations in first mention please.

·

Basic reporting

No comments

Experimental design

Methods could have been more elaborate so that replication of information can be made more realistic and achievable. For instance, what was the methodology involved in HDL-C estimation?

Validity of the findings

1.Findings could have been made more logical and convincing rather than merely striking a relationship with HDL-C
2. Earlier studies have reported the association/nexus with cardiometabolic factors. The authors could have compared the levels of METRNL with adiponectin, for instance, with reference to HDL-C status
3.HDL-C per se may not be linked to insulin resistance. Thee authors should have taken into consideration factors such as QUICKI, atherogenic index of plasma and cardiac risk index based on lipid ratios, including the surrogate marker for insulin resistance, namely Triglyceride:HDL ratio
4.BMI could have been used to segregate the study subjects into non-obese, overweight and obese which could have thrown more weight on the anthropometric measurement
5. Mean arterial Pressure could have been used instead of SBP and DBP

Additional comments

Spelling and grammatical errors noted. For e.g. instead of the correct spelling- comparison, the authors have used comparition in a few places

---

## Round 0.2 · accepted · Accept

Congratulations.

Yours,

Yoshi
Prof. Yoshinori Marunaka, M.D., Ph.D.

Reviewer 1 ·

Basic reporting

The paper is well revised and significantly improved.

Experimental design

Study design is well and suitable to answer the study question.

Validity of the findings

Findings are valid

Additional comments

Revisions are adequate and the paper is improved.

·

Basic reporting

Adequate and replete with the necessary information.

Experimental design

Substantiated appropriately with suitable justification.

Validity of the findings

Original contributions to the world of evidence based research is obvious and the authors have enhanced the quality of the findings by taking due cognizance of the comments and suggestions of the Reviewer.

Additional comments

No comment